# Ambient Climate Influences Anti-Adhesion between Biomimetic Structured Foil and Nanofibers

**DOI:** 10.3390/nano11123222

**Published:** 2021-11-27

**Authors:** Marco Meyer, Gerda Buchberger, Johannes Heitz, Dariya Baiko, Anna-Christin Joel

**Affiliations:** 1Institute for Biology II, RWTH Aachen, Worringerweg 3, 52074 Aachen, Germany; 2Institute of Applied Physics, Johannes Kepler University Linz, Altenberger Strasse 69, 4040 Linz, Austria; gerda.buchberger@jku.at (G.B.); johannes.heitz@jku.at (J.H.); 3Institute of Biomedical Mechatronics, Johannes Kepler University Linz, Altenberger Strasse 69, 4040 Linz, Austria; dariya.baiko@jku.at

**Keywords:** spider silk, calamistrum, LIPSS, humidity, temperature, van der Waals, bionic, laser structuring, fiber processing, nanofibers

## Abstract

Due to their uniquely high surface-to-volume ratio, nanofibers are a desired material for various technical applications. However, this surface-to-volume ratio also makes processing difficult as van der Waals forces cause nanofibers to adhere to virtually any surface. The cribellate spider *Uloborus plumipes* represents a biomimetic paragon for this problem: these spiders integrate thousands of nanofibers into their adhesive capture threads. A comb on their hindmost legs, termed calamistrum, enables the spiders to process the nanofibers without adhering to them. This anti-adhesion is due to a rippled nanotopography on the calamistrum. Via laser-induced periodic surface structures (LIPSS), these nanostructures can be recreated on artificial surfaces, mimicking the non-stickiness of the calamistrum. In order to advance the technical implementation of these biomimetic structured foils, we investigated how climatic conditions influence the anti-adhesive performance of our surfaces. Although anti-adhesion worked well at low and high humidity, technical implementations should nevertheless be air-conditioned to regulate temperature: we observed no pronounced anti-adhesive effect at temperatures above 30 °C. This alteration between anti-adhesion and adhesion could be deployed as a temperature-sensitive switch, allowing to swap between sticking and not sticking to nanofibers. This would make handling even easier.

## 1. Introduction

Nanofibers promise immense opportunities for innovation. Already today they are integrated into efficient filters, smart textiles, and biomedical applications [1,2,3,4,5]. However, their technical application is limited, as their processing is difficult. Due to their uniquely high surface-to-volume ratio, which is very advantageous in many of their applications, nanofibers adhere to virtually any surface on account of the prevailing van der Waals forces [6,7]. These forces are the molecular forces between (induced) dipoles forming at surfaces in very close proximity [8]. One famous example of the surprising strength of van der Waals forces when deploying nanostructures are the sticky gecko feet, adhering the gecko to the ceiling [9,10]. To learn how to process nanofibers though they stick to every surface, one can look again at nature, where some spiders can spin complex threads with thousands of nanofibers without sticking to these [11].

These so-called cribellate spiders produce thousands of nanofibers as adhesive parts of their capture threads (Figure 1a,b). The 10 to 30 nm thick nanofibers are wrapped around supporting axial fibers and, depending on the species, further silk types can be involved, influencing the mechanical properties of the thread [12,13,14,15,16,17]. The spiders extract the nanofibers from spigots of the eponymous cribellum, a spinning plate on the abdomen of cribellate spiders [18,19]. During the extraction, the spiders use a specialized comb, the calamistrum, on their hindmost (fourth) leg (Figure 1c). The calamistrum is not necessary for the extraction of the nanofibers themselves, but thread structure differs largely between combed and uncombed threads [15,20]. Despite the direct contact with the nanofibers during combing, the calamistrum remains free of fiber residues [21,22]. Correct contact seems essential for the functionality of the calamistrum [23].

Previous studies demonstrated that the calamistra of the feather-legged lace weavers, *Uloborus plumipes*, are equipped with nanoripples [21,24] (Figure 2a,b). These ripples minimize contact between nanofibers and calamistrum, as the fibers rather bridge over the gaps between two ripples than bend and adapt to the surface topography. As a result, van der Waals forces are reduced [21]. This functionality can be transferred to artificial surfaces: Joel et al. [21] demonstrated the biomimetic potential by transferring the spider’s nanostructure to poly(ethylene terephthalate) (PET) foils using a pulsed nanosecond KrF* laser with a wavelength *λ* of 248 nm (Figure 2c). This resulted in the self-organized formation of nanoripples, namely in laser-induced periodic surface structures (LIPSS). In general, nanoscale structures on surfaces can be manufactured by different laser processing methods. LIPSS are among the typically applied ones [22,25,26]. LIPSS form on solid or liquid surfaces after laser irradiation with polarized light within a certain range of laser processing parameters [27]. Following the most basic theory regarding their generation, they are associated with the optical interference of the scattered laser beam with the incident laser beam on surfaces with nanoscale roughness [28]. For LIPSS formation on polymers such as flexible PET foils [29,30,31,32], typically thousands of nanosecond laser pulses have to be applied at the same irradiated spot. Hereby, the used fluence must be kept well below the single-pulse ablation threshold. The resulting ripples are usually oriented parallel to the direction of the linear polarization. In case of s-polarized laser light with a wavelength *λ*, the spatial period Λ of the LIPSS on polymers is given by the formula Λ = *λ/*(*n*_eff_ − sin*θ*), where *n*_eff_ is the effective refractive index and *θ* the angle of incidence of the laser beam. Hereby, *n*_eff_ lies between the refractive index of air (≈1) and the refractive index of the polymer. The height of the ripples *h* cannot be chosen independently from Λ. LIPSS formed on PET showed anti-adhesive properties comparable to the biological role model and reduced adhesion by ~40% [21,22].

Having a surface with non-sticky properties towards nanofibers is exciting, as this enables easier processing, such as spooling of nanofibers. However, van der Waals forces are known to be influenced by temperature and humidity [33]. Nanofiber production is often air-conditioned; if the biomimetic surfaces turn out to be sensitive to climatic conditions, potential boundaries and conflicts must be explored before incorporation into larger industrial machinery. This study is the first to pin down boundary conditions for the performance of biomimetic surfaces for nanofiber handling. It aims to advance the industrial exploitation of the anti-adhesive ripples of the calamistrum and thus to facilitate the technical processing of nanofibers. To this end, we explored the influence of relative humidity (RH) and temperature on the adhesion-reducing performance of LIPSS equipped PET foils. Since cribellate spiders can be found in a wide variety of habitats, ranging from hot to cooler regions and dry to humid environments, we do not expect climatic conditions to influence the anti-adhesion of our biomimetic surfaces. However, gecko adhesion, a natural example also based on van der Waals forces, is known to be hampered by humidity and temperature [34,35].

For this study, we used the same combination of natural spider nanofibers and PET samples, as described in the study demonstrating this effect for the first time [21]. To this end, we have also chosen the same laser processing setup and fabrication parameters for the biomimetic PET foils as in [21] as well as the same foil thickness, namely 50 μm, and type, i.e., biaxially stretched foils. Hence, our data should be comparable to the data presented before [21]. Although the spider’s nanofibers have a much smaller diameter than typical technically produced nanofibers (~20 nm compared to a few hundred nm), it is generally aimed to scale down the diameter of technical fibers, making handling more and more complex [36,37].

## 2. Materials and Methods

Ethics. The species used in the experiments are not endangered or protected. Special permits were not required. All applicable international, national, and institutional guidelines for the care and use of animals were followed.

Study animals. Adult *Uloborus plumipes* (Lucas, 1846) of similar size were caught in garden centers across Aachen (Germany). They were held separately in 11 cm × 11 cm × 6.5 cm boxes with roughened surfaces, where they could build their webs under room temperature, room relative humidity, and central European diurnal rhythm in the laboratory. The spiders were fed weekly with *Drosophila melanogaster* and water was provided via soaked cotton balls biweekly. Capture threads were collected from the webs before each experiment without stretching them. To ensure equal thread length, threads were placed on a 7.7 mm wide sample holder. Double-sided sticky tape was used to prevent the threads from being displaced. It is known that the puff structures collapse upon wetting and thus the adhesion drops [38,39]. The threads were therefore examined by light microscopy before and after the experiments to discard ruined threads. However, we never observed collapsing threads, also at higher humidity levels, which means collected threads had no contact with fluid water droplets [40].

Production of structured foils. Flat, biaxially stretched PET foils with a thickness of 50 μm (Goodfellow Ltd., Huntingdon, UK) served as a substrate for laser-induced periodic surface structures (LIPSS) mimicking the ripple structures on the calamistrum. A KrF* excimer laser (LPX 300, Lambda Physik, Göttingen, Germany) was used (Figure 3), generating nanosecond UV light pulses with τ ≈ 20 ns (duration) and *λ* = 248 nm (wavelength). For sample fabrication, a pulse repetition frequency of 10 Hz and *N* = 6000 pulses with a fluence of approximately 8.1–9.2 mJ/cm^2^ were chosen, because these parameters result in regular ripples on the whole processed area. Lower fluences of approximately 4 mJ/cm^2^ result in nanodots and not in nanoripples. At fluences higher than the applied ones, e.g., of about 13 mJ/cm^2^, microstructures are formed in addition to the nanoripples. So far, we have not observed a notable effect of the fluence on their heights in this range. An α-BBO polarizer (Melles Griot, Carlsbad, CA, USA) generated linearly polarized light. This output was then imaged onto the samples by two fused silica lenses in a telescope configuration. A high-power variable attenuator with a dielectric coating (magnetron sputtered and with anti-reflection layer, Laseroptik GmbH, Garbsen, Germany) enabled beam energy adjustment by being mounted on a rotatable stepper motor. We measured beam energy with a pyroelectric joulemeter (Gentec from Soliton Laser-und Messtechnik GmbH, Gilching, Germany, model “ED-500”); this was placed after the last lens. With the help of a rotatable sample holder, the laser hit the surface at *θ* = 30° (angle of incidence). The resulting processed area is elliptic and about 1.3 cm^2^ large.

Non-contact AFM. Atomic force microscope (AFM) images were obtained from PET foils sputter-coated with gold (Bal-Tec CD005; Balzer, Liechtenstein) for 80 s resulting in a 10–20 nm thick gold layer. The foil was sputter-coated in bent state to avoid possible cracking and/or stretching of the gold layer, as it needs to be looped for adhesion measurements. A Digital Instruments CP II set-up (Veeco, Munich, Germany) was operated in the non-contact mode (rectangular tip RTESPA-300 from Bruker (Karlsruhe, Germany) with nominal/maximal radii of 8/12 nm, minimal/nominal/maximal frequencies of 200/300/400 kHz, minimal/nominal/maximal lengths of 115/125/135 µm, minimal/nominal/maximal widths of 38/40/42 µm and minimal/nominal/maximal spring constants of 20/40/80 N/m). The tip was moved approximately vertically to the ripples on the surface for image generation. AFM data was leveled by mean plane subtraction and the minimum data value was shifted to zero by the respective functions in the free software Gwyddion (Version 2.55, Czech Metrology Institute, Brno, Czech Republic). In our data, significant convolution of the surface morphology with the geometry of the AFM tip occurred. Therefore, smaller features and especially the heights of the ripples are not resolved properly.

Anti-adhesion experiments. We tested structured (LIPSS) and unstructured (control) foils. To ensure edgeless contact and thus prevent tangling of the fibers, the foils were cut into 1.5 cm long strips and clamped in a loop form. We took caution not to kink the foils. To account for laser-induced chemical surface modification, pristine and laser-processed PET foils were sputter-coated (Hummer Technics Inc., Alexandria, VA, USA) with an approximately 10 nm thick gold layer (5 min with 7.5 mA, 1000 V) to ensure equal surface chemistry between control and structured PET as well as to reduce electrostatic charging of the PET foils during the experiments. Nanosecond KrF* excimer laser processing does not only change the surface topography of PET foils but also their surface chemistry in a complex manner [32,41,42,43]. Roughly speaking, depending on the applied laser fluence, they may become either more hydrophilic or hydrophobic compared to pristine foils due to changes in the carbon versus oxygen ratios of the surface; this was revealed by X-ray photoelectron spectroscopy (XPS) [32,41,42,43], attenuated total reflection Fourier transform infrared spectroscopy (ATR-FTIR) and static contact angle measurements [41,43]. Using angle-resolved XPS, Siegel et al. [32] observed a different chemical composition between the valleys and ridges of nanoripples for a fluence of 6.6 mJ/cm^2^, whereas Richter et al. [41] did not measure such differences for a fluence of 5.7–6.2 mJ/cm^2^. To account for laser-induced chemical surface modification, pristine and laser-processed PET foils were sputter-coated with gold in the previous study of Joel et al. [21], when the effect of nanoripples on anti-adhesion was studied without the influence of altered surface chemistry.

In a climate chamber (HPPeco; Memmert GmbH & Co. KG, Schwabach, Germany), a motorized micromanipulator (MT30-50; Standa Ltd., Vilnius, Lithuania) was used to bring prior acclimatized thread samples into contact with foil samples. A small deformation of the thread confirmed contact, whereupon the thread was slowly and constantly (*v* = 2 mm/s) withdrawn from the foil until it detached. Via a video recording microscope (VW-9000C; Keyence Corporation, Osaka, Japan), the deflection was recorded at 10 to 20 × magnification and 60 fps. To minimize electrostatics, samples were grounded prior to trials and, if necessary, the air was ionized via an ion-gun (Milty Zerostat 3; Armour Home Electronics Ltd., Bishop’s Stortford, UK). Employing the climatic chamber, measurements were conducted at relative humidity (RH) levels of 10%, 30%, 50%, 70%, and 90% (with *T* = 20 °C) and temperatures of 10 °C, 20 °C, 30 °C and 40 °C (with RH = 50%). Per condition, 25 structured (LIPSS) foils as well as 25 unstructured (control) foils were tested. Threads and foils were used only once. The experiments were shuffled and carried out over several days. Data were excluded if entanglement or excessive electrostatic effects were visible before or during the trial and additional trials were performed to keep the sample size constant. Adhesion was determined by measuring the maximum deflection perpendicular to the initial thread position using the Keyence VW-9000 motion analyzer software (version 1.4.0.0, Keyence Corporation, Osaka, Japan).

Statistical analysis. Data are presented as mean ± standard deviation (SD). Calculations and analysis were performed in R via RStudio integrated development environment (Version 1.4.1717, R Foundation for Statistical Computing, Boston, MA, USA) [44]. After normal distribution was confirmed by means of a histogram, we used an ANOVA with subsequent Tukey post-hoc test to compare the individual treatments with each other. To this end, we assumed independent observations for all experiments. A *t*-test (Wilcoxon) was performed to compare LIPSS and control data individually. Significance was assumed if *p* < 0.05.

## 3. Results

Biomimetic rippled PET foils can mimic the anti-adhesive properties of the calamistrum towards nanofibers well [21]. This is promising for processing technical nanofibers, as these fibers tend to stick to all surfaces by van der Waals forces. To explore the boundary conditions under which this fiber handling can be performed in the future, we tested different relative humidity levels and temperatures and their influence on the anti-adhesive properties of the nanorippled surfaces. In our experiments, ripples had a spatial period of about 320 nm (Figure 4). As the height is linked to periodicity and we used the same processing settings as Joel et al. [21], we assume the ripples had a similar height of about 100 nm.

### 3.1. Influence of Temperature and Humidity on the Mechanical Properties of the Cribellate Thread

As we measured the forces indirectly (i.e., the displacement of the thread) we performed all experiments with a corresponding non-structured sample (controls). We suspected a changed mechanical performance of the thread, as a changed plasticity was shown for other silks at high humidity [45]. We observed a greater deflection of the threads only at 90% humidity (*p* < 0.05, ANOVA with post-hoc Tukey test) and no significant influence of temperature (Table 1 and Table 2). However, with our data, we cannot discriminate whether the difference at 90% humidity was due to increased adhesion to the surface or changed plasticity of the thread.

### 3.2. Response to Changing Ambient Climate

At low and high humidity, i.e., below 50% and above 90% RH, LIPSS could reduce the adhesion between nanofibers and foils by almost 50% (Table 1, Figure 5a). At 70%, however, the nanofibers adhered equally well to both surfaces, the control, and LIPSS foil.

All humidity experiments were performed at 20 °C. At lower temperatures, i.e., 10 °C, the adhesion towards the LIPSS foil was reduced by about 30%. However, when increasing the temperature (i.e., 30 °C and 40 °C), the threads showed increased adhesion to LIPSS foils and the adhesion of nanofibers to LIPSS foils and controls did not differ anymore (Table 2, Figure 5b).

## 4. Discussion

Nanofibers are very advantageous for many applications, for example, filter membrane production. However, their production and processing are both challenging, with one problem being their stickiness due to van der Waals forces. A previous study showed that biomimetic foils covered with a nanoripple structure can reduce the adhesion between foil and nanofibers (Figure 6) [21]. With this study, we show that although there are few limitations regarding the ambient humidity, temperature can influence the anti-adhesive properties of the biomimetic foil. This temperature effect could serve as a switch between adhesion and anti-adhesion, which could, in fact, be useful in the processing of nanofibers.

### 4.1. Influence of Humidity on Anti-Adhesion

Consistent with gecko adhesion, nanofiber adhesion increased at high humidity [35]. This effect has already previously been shown for cribellate fibers [46,47]. However, we did not observe a consistent influence of humidity on the anti-adhesive properties of the LIPSS structured foil. Anti-adhesion provided by the LIPSS could be demonstrated at both low and high humidity levels. The effect was most pronounced at 50% RH. Only at 70% RH, we were not able to detect a difference between controls and our LIPSS structured foils. Given the high standard deviation, however, this could be due to a variance of the mechanical properties of the silk fibers used as well as irregularities of the LIPSS. The plasticity of spider silk can increase at higher humidity levels as water molecules disrupt the hydrogen bonds between protein chains, causing the elastic modulus to decrease and the fibers to become more compliant [45,48]. If these results are transferrable to our silk nanofibers, nanofibers with increased compliance could better adapt to the ripple structure of the LIPSS, resulting in a larger contact surface between fiber and foil and, consequently, an increase of van der Waals forces [16,21]. However, this should also lead to an increased adhesion at a humidity of 90%. On the other hand, capillary condensation in the nanometer-wide valleys of the LIPSS could occur at high humidity [47,49,50]. The hydrophilicity of the gold layer would promote such condensation [51]. Again, this should also lead to higher adhesion at 90%, which is not observable.

In the controls, fiber deformation increased only at 90% humidity, whereas it seems in the LIPSS structured foils fiber deformation increases already at 70% humidity and does not change further when increasing the humidity above. Hence, we suspect a transition level where mechanical properties already show initial changes [52] and also early hygroscopic effects in the valleys are conceivable. As both factors depend on fiber properties or surface chemistry, respectively, the performance for the specific fiber/surface combination in technical applications operating at this humidity level should be verified beforehand. In general, lower humidity seems to be most beneficial for minimizing adhesion. This is particularly interesting as low humidity is advantageous for the production of nanofibers and has a positive effect on the fiber diameter [53,54]. The ideal relative humidity for electrospinning of commonly used polymers seems to be at 30% [55].

### 4.2. Influence of Temperature on Anti-Adhesion

At typical room temperature and below, the anti-adhesive effect of LIPSS is pronounced. However, when increasing temperature to 30 °C and above, anti-adhesion could no longer be observed. High temperatures are known to potentiate the influence of humidity on silk fibers [45]. Thereby, the required humidity level at which the elastic modulus declines is greatly reduced. However, we consider such an effect on the nanofibers to be unlikely since spider dragline silk does not transition to a rubber state until 80 °C [45]. Furthermore, we found no temperature-related increase of deflection on unstructured control surfaces. Therefore, we suspect an effect exerted on the LIPSS themselves that influences van der Waals forces.

The non-reduced adhesion at 30 °C and higher does not necessarily represent a drawback: The change from anti-adhesion to adhesion and vice versa, by simply changing temperature could enable easier handling of nanofibers. For example, the setup could be slightly heated for spooling and then easily further unwind and processed after cooling.

## 5. Conclusions

With this study, we demonstrated that biomimetic LIPSS can significantly reduce the adhesion of 10 to 30 nm thick fibers, especially at common laboratory conditions. Surprisingly, we found temperature to be a limiting factor, rather than humidity: at an ambient temperature of 30 °C, the behavior switched from anti-adhesive to normal adhesion. Choosing the most favorable of the tested ambient conditions, the reduction of fiber deflection could be further increased by about 6% (from ≈−40% to ≈−46%) compared to the previous study, where the ambient conditions were not accounted for [21]. Furthermore, in this work, we changed only one of the two parameters at a time, while the other was held constant at a predefined level. In case of a humidity dependency of the temperature effect, lower humidity might increase the temperature tolerance and the anti-adhesion could be achieved even in warmer environments, provided humidity is kept low. Furthermore, we strongly expect, that the measured climatic limitations can be overcome: Cribellate spiders, such as our model organism *U. plumipes*, can be found in various habitats, ranging from warm and dry to humid habitats [56]. They build their webs under the prevailing conditions and consequently, we expect that the described restrictions can be overcome by further adapting the LIPSS to the requirements. The LIPSS produced for this study differ from the natural nanostructures on the calamistrum in their spatial frequency but also in the depth of the valleys between the ripples [21,22,24]. To this end, tailoring the ripples to the appropriate fibers and their mechanical properties in the desired ambient climate could lead to anti-adhesive behavior at a wider range of humidity and temperature.

## Figures and Tables

**Figure 1 nanomaterials-11-03222-f001:**
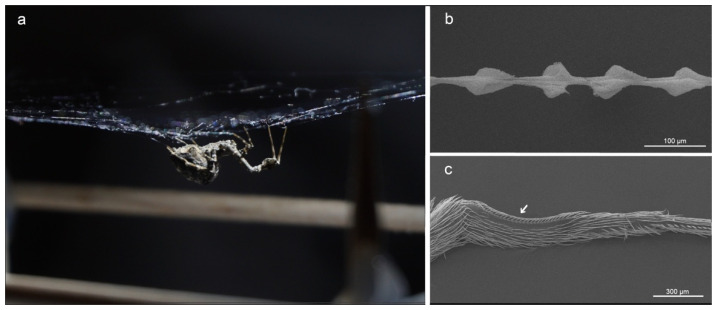
Cribellate spiders use combs on their hindlegs to brush nanofibers in puffy structures. (**a**) The feather-legged lace weaver, *Uloborus plumipes*. (**b**) SEM images of the spider’s capture thread employing a mesh of thousands of nanofibers forming puffy structures. (**c**) SEM images of the hind leg of *U. plumipes*. The arrow indicates the position of the calamistrum, the anti-adhesive comb on the metatarsus. Specimens were gold-sputtered before SEM examination.

**Figure 2 nanomaterials-11-03222-f002:**
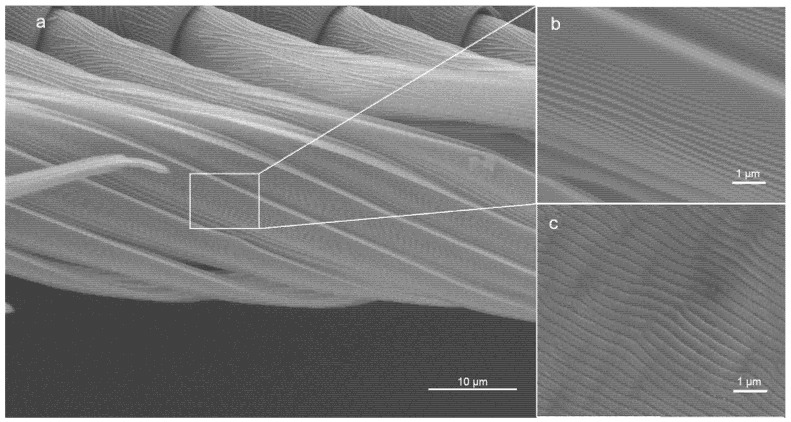
The nanostructures on the calamistrum were biomimetically replicated on PET via a nanosecond UV laser. (**a**) The calamistrum of the feather-legged lace weaver. (**b**) Nanoripples on the calamistrum. (**c**) LIPSS on PET foil. SEM images of gold-sputtered specimens.

**Figure 3 nanomaterials-11-03222-f003:**
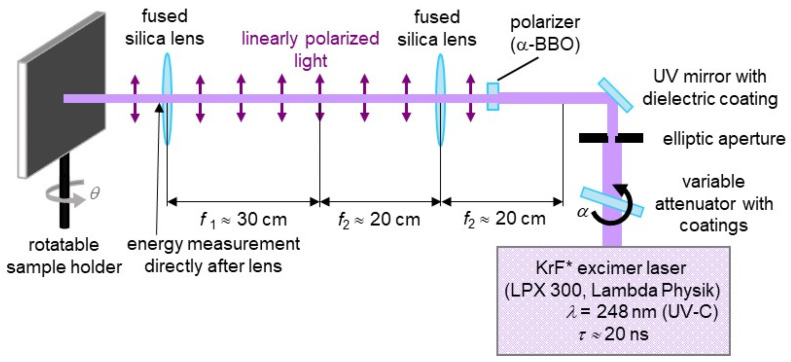
Schematic of the used KrF* excimer laser setup for the production of nanoripples on PET foils. Adapted from ref. [41].

**Figure 4 nanomaterials-11-03222-f004:**
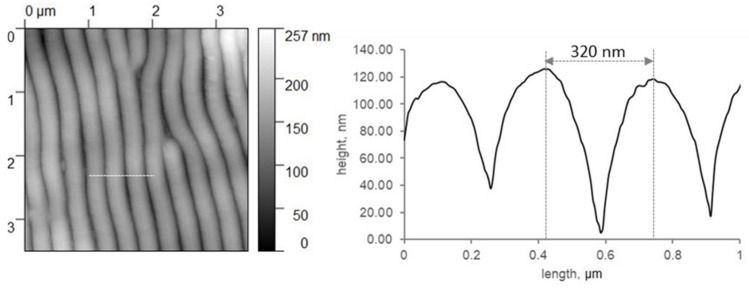
Non-contact AFM data to characterize the rippled LIPSS of the PET foils. The right graph shows the profile along the white line in the topography image on the left.

**Figure 5 nanomaterials-11-03222-f005:**
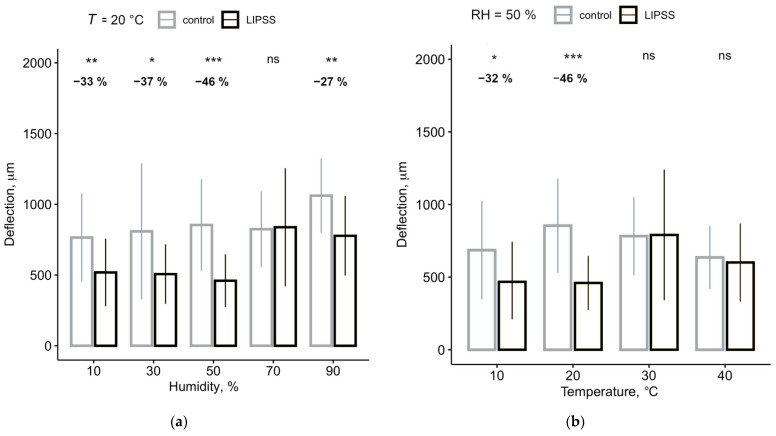
The anti-adhesive effect is most effective in laboratory conditions. Shown are the averaged maximum deflections of the threads before detaching from the surface. In the area above, the difference of the deflections between control and structured PET (in %) is indicated as well as the statistical relevance. (**a**) Comparison of LIPSS and control foil at different relative humidity levels but constant temperature (*T* = 20 °C); (**b**) Comparison of LIPSS and control foil at different temperatures but constant humidity (RH = 50%). *n* = 25 for (a) and (b). Significant differences between LIPSS and control foils were determined by a Wilcoxon *t*-test. *: *p* < 0.05, **: *p* < 0.01, ***: *p* < 0.001, ns: not significant.

**Figure 6 nanomaterials-11-03222-f006:**
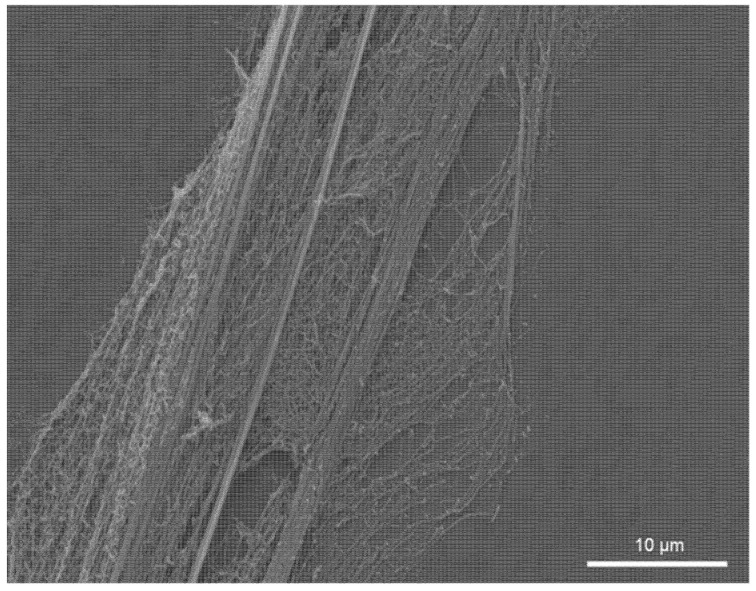
SEM image of nanofibers on a biomimetic polymer foil, gold-sputtered after interaction. A cribellate capture thread with its nanofibers lies perpendicular to the LIPSS. These prevent the thread from smoothly adapting to the surface and thus reduce adhesion.

**Table 1 nanomaterials-11-03222-t001:** Indirect measurement of the adhesion (by deflection) of the threads at different humidity levels, presented as mean ± SD and the relative change between control and structured surface (Diff.). Significance was calculated using a Wilcoxon *t*-test, with *p* < 0.05.

Humidity	Deflection		*p*-Value	Diff.
	Control PET	LIPSS PET		
10%	765 ± 312 µm	518 ± 211 µm	0.0036	−32.63%
30%	809 ± 482 µm	506 ± 239 µm	0.0137	−37.36%
50%	854 ± 323 µm	459 ± 187 µm	3.5 × 10^−6^	−46.18%
70%	824 ± 268 µm	838 ± 418 µm	ns.	+1.77%
90%	1061 ± 263 µm	777 ± 282 µm	0.0013	−26.75%

**Table 2 nanomaterials-11-03222-t002:** Indirect measurement of the adhesion (by deflection) of the threads at different temperature levels, presented as mean ± SD and the relative change between control and structured surface (Diff.). Significance was calculated using a Wilcoxon *t*-test, with *p* < 0.05.

Temperature	Deflection		*p*-Value	Diff.
	Control	LIPSS		
10 °C	685 ± 337 µm	468 ± 277 µm	0.023	−31.73%
20 °C	854 ± 323 µm	460 ± 286 µm	3.5 × 10^−6^	−46.18%
30 °C	782 ± 269 µm	791 ± 460 µm	ns.	+1.14%
40 °C	636 ± 217 µm	600 ± 270 µm	ns.	−5.59%

## Data Availability

All data is included in this manuscript.

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
