# Peer review of "Ambient Climate Influences Anti-Adhesion between Biomimetic Structured Foil and Nanofibers"

_nanomaterials, 2021, doi:10.3390/nano11123222_

Round 1

Reviewer 1 Report

Meyer et al. present an interesting and valuable study of influences of temperature and humidity upon engineered substrate-cribellate silk fibre interactions. Namely, laser-induced periodic surface structures were produced on PET films, coated with gold and then tested for their adhesive behaviour with respect to cribellate silks. Surface structuring was evaluated by AFM and adhesion to either surface structured or control substrates in an environmental control chamber by videomicroscopic measurement of maximum deflection perpendicular to the thread position. As a whole, a convincing and robust set of data are provided, showing a temperature dependence in anti-adhesive behaviour of the surface structured substrate (i.e., a loss of anti-adhesive behaviour above 30 °C), without a clear relation of anti-adhesive behaviour to relative humidity. Notwithstanding my general overall enthusiasm for the work, I do have a few concerns and suggestions for revision:

1) Most importantly, the AFM imaging results shown appear to be demonstrating significant tip convolution in the cross-sectional height trace. (Namely, the “troughs” that are shown appear to correspond to a limitation in the ability of the AFM probe tip to reach any further into the trench.) This would ideally be rectified through use of sharper probes with a steeper cone angle than the RTESPA-300 probes that were used. Alternatively, this limitation could be acknowledged with a focus strictly on the lateral periodicity of the surface features and removal of quantification of/discussion of depths. It should also be noted that the x-axis units in the right-hand panel are incorrect.

2) The SD values for deflection appear quite large overall. Was any statistical outlier testing used (over and above the note that certain data were excluded)? If not, would the data be amenable to clean up through evaluation of statistical outliers? I recognize that statistical testing was applied for significance, but some of the differences that are noted as being “significant” are a bit hard to rationalize as being truly different based on examination of Figure 5. Note also that in Fig. 5, the noted “-32%” in the left panel should actually be “-33%”.

3) On p. 4 (lines 127-129), gold coating is mentioned. Additional detail of this process would be beneficial – i.e., what thickness of gold is applied? Also, what type(s) of altered surface chemistry are anticipated and are there some prior studies that demonstrate that this is an effective means to alleviate these? (Presumably, if this was strictly a surface charging issue, dramatically changing the surface through coating in this manner would not be required to rectify this issue.)

Some minor notes:

Line 10 “is also making” should be “also makes”

Line 31 – consider rewording “their special surface-to-volume ratio”, or at least expanding upon this

Line 32 – “applications, the nanofibers” seems more correct as “applications, nanofibers” as at this point generic nanofibers are being discussed, rather than a specific type of nanofiber.

Lines 227-232 – this logic is a little hard to follow, given the outlier nature of the data at 70% RH. It may be beneficial to reword/change this discussion a little for clarity.

Line 247 – “depend on fiber respectively surface chemistry” needs to be corrected/reworded

Line 276 – period missing

Lines 286-287 – the personal observation makes sense, but it may be more appropriate to word this in a more descriptive manner vs. treating this in the same manner as a reference without attribution to the person(s) who are making the observation.

Author Response

Meyer et al. present an interesting and valuable study of influences of temperature and humidity upon engineered substrate-cribellate silk fibre interactions. Namely, laser-induced periodic surface structures were produced on PET films, coated with gold and then tested for their adhesive behaviour with respect to cribellate silks. Surface structuring was evaluated by AFM and adhesion to either surface structured or control substrates in an environmental control chamber by videomicroscopic measurement of maximum deflection perpendicular to the thread position. As a whole, a convincing and robust set of data are provided, showing a temperature dependence in anti-adhesive behaviour of the surface structured substrate (i.e., a loss of anti-adhesive behaviour above 30 °C), without a clear relation of anti-adhesive behaviour to relative humidity. Notwithstanding my general overall enthusiasm for the work, I do have a few concerns and suggestions for revision:

Dear reviewer,

Thank you for taking the time to review our manuscript. We appreciate your kind words and are happy you liked our work and that our intention was well understood. Below you find our replies to your valuable comments.

1) Most importantly, the AFM imaging results shown appear to be demonstrating significant tip convolution in the cross-sectional height trace. (Namely, the “troughs” that are shown appear to correspond to a limitation in the ability of the AFM probe tip to reach any further into the trench.) This would ideally be rectified through use of sharper probes with a steeper cone angle than the RTESPA-300 probes that were used. Alternatively, this limitation could be acknowledged with a focus strictly on the lateral periodicity of the surface features and removal of quantification of/discussion of depths. It should also be noted that the x-axis units in the right-hand panel are incorrect.

We thank you very much that for pointing out this very important issue. For sure, there is a need to discuss it in our paper. We followed the suggestions and removed the mean ripple heights which were evaluated from AFM topographies, where convolution occurred. Unfortunately, we were not able to record new data of LIPSS sputter-coated with gold using another, sharper tip on such short notice. This is also due to the fact, that the used AFM is currently out of order and needs to be repaired or replaced. We would be happy to add new data of the depth of the ripples, testing also other methods (such as FIB-SEM) to evaluate these. However, we would need a longer revision time for such procedure. Hence, we followed your suggestions and removed the mean ripple heights. As we used the same parameters as in the study before, we assume the depth to be similar (added as information also to the results). Furthermore, we discussed the convolution between the surface morphology and the geometry of the AFM tip (see material and methods).

We have changed the unit of the x-axis, thank you for pointing this out.

2) The SD values for deflection appear quite large overall. Was any statistical outlier testing used (over and above the note that certain data were excluded)? If not, would the data be amenable to clean up through evaluation of statistical outliers? I recognize that statistical testing was applied for significance, but some of the differences that are noted as being “significant” are a bit hard to rationalize as being truly different based on examination of Figure 5. Note also that in Fig. 5, the noted “-32%” in the left panel should actually be “-33%”.

During data evaluation we also noted the high SD. That’s why we already removed outlier (see Material & methods for electrical charging problem). We have again searched for statistical outliers, but our data span the entire range (see plots below). Furthermore, there is no other effect than electrical charging or fiber entanglement that would justify a removal. We suspect the high SD due to A) irregularities of LIPPS (Bonse et al. 2017) as well as B) biological variance within the silk samples (for example, age and food of the spiders, i.e., Blamires et al., 2014). We have now addressed this issue in the discussion of the manuscript.

We have corrected figure 5.

3) On p. 4 (lines 127-129), gold coating is mentioned. Additional detail of this process would be beneficial – i.e., what thickness of gold is applied? Also, what type(s) of altered surface chemistry are anticipated and are there some prior studies that demonstrate that this is an effective means to alleviate these? (Presumably, if this was strictly a surface charging issue, dramatically changing the surface through coating in this manner would not be required to rectify this issue.)

We have added additional details about the gold coating and the coater used. In the material section The approx. thickness is also now noted in the method section.

Additionally, we changed the wording regarding the surface chemistry. Surface chemistry can be changed by laser processing (Bonse et al., 2000); a paragraph about surface chemistry alteration by KrF* excimer laser treatment is now also included in the materials and methods). As we did not characterize the surface chemistry of our samples and additionally electric charging was a major problem in our experiments (not because it exists, but it is hardly controllable for its consistency between experiments, if not trying to reduce it as much as possible), we chose the gold coating by sputter coating. Thank you, for pointing out this missing issue

Some minor notes:

Line 10 “is also making” should be “also makes”

Line 31 – consider rewording “their special surface-to-volume ratio”, or at least expanding upon this

Line 32 – “applications, the nanofibers” seems more correct as “applications, nanofibers” as at this point generic nanofibers are being discussed, rather than a specific type of nanofiber.

We have implemented all linguistic suggestions and revised the complete manuscript in this respect. Due to a short revision deadline of 7 days, we were not able to present the manuscript to a native speaker.

Reviewer 2 Report

All comments are included in attached file.

Author Response

The submitted paper is focused on laser-induced treatment of polyethylene terephthalate to mimic non-stickiness of the calamistrum by fabrication of periodic surface structures on the PET substrate. Climatic conditions, such as temperature and humidity, influence on the anti-adhesive performance of fabricated biomimetic surfaces was investigated in this study. SEM and AFM microscopic techniques were implemented in this study to analyze surface morphology and topography of fabricated surfaces. The presented results are consistent having a potential use for application in anti-adhesion applications in polymer field. Submitted paper is recommended for publication in this journal after a minor revision.

Dear reviewer,

Thank you for taking the time to review our work and especially for recommending our submitted paper for publication. We have taken into account all your comments and suggestions, please see a detailed point-to-point answer to your questions below.

In the introduction, the novelty and potential applications of this research is missing. More information about laser treatment could be provided.

Thank you for pointing this out. We revised the introduction accordingly to emphasize our novelty and targeted application. We have additionally included further information about laser treatment into the introduction. We did not include more information on laser treatment in the initial manuscript because we wanted to put the emphasis on the (anti-) adhesion under different ambient conditions, although we understand now that important information was missing for the reader

In the Experimental section, information about used AFM tip would be useful. Why these laser parameters were used for the surface treatment of PET, is was based on previous optimization? Have authors analyzed the effect of laser intensity on ripples formation?

The respective information (AFM and why choosing these laser parameters) was included into the materials and methods. Please note that the AFM data was revised according to a suggesting of the other reviewer.

In the Results section, 3D images from AFM could be included in Figure 4. Figure 6 could be described detailed. Symbols in Table 1 and Table 2 should be described detailed, such as deflection or p-value.

You are right: we have not explained this topic well enough in our initial manuscript. Therefore, we have added the first part of the respective explanation in the last paragraph of the introduction and the second part in the subsection “Production of structured foils” in materials and methods. We have added more information to the figure 6 and adjusted the table descriptions.

In conclusions section, full stop is missing in “We 274 found surprisingly temperature to be a limiting factor, rather than humidity: at an ambi-275 ent temperature of 30 °C the behavior switched from anti-adhesive to adhesive Choosing”. Please check this sentence as well “compared to the pre-278 vious study, where the ambient conditions were not accounted for”.

We have adopted all your linguistic comments. In addition, we have revised the complete text again with regard to the language (see also comments of reviewer 1 and editor).

Round 2

Reviewer 1 Report

Whilst I would have preferred to see the authors given time to address my concern around the AFM methodology and resulting analysis in the first round of revision, the decision by the journal that the revised manuscript had to be submitted within 7 days did not allow for this. This is not the fault of the authors, and I am willing to give them the benefit of the doubt on the comparability of the current methodology for substrate processing to their related prior work, as now detailed in the revision. (Ideally, the authors would have been given sufficient time in revision to either carry out additional experiments or to provide the clear rationale that they have for their substrate properties, rather than being forced to do the latter.) Beyond this issue, I feel that my comments and concerns from the first round of review have been satisfactorily and comprehensively addressed.

Reviewer 2 Report

Authors considered almost all my comments and suggestions adn therefore I recommend this article in current form for publication in this journal.